# The Role of Fibrogenesis and Extracellular Matrix Proteins in the Pathogenesis of Graves’ Ophthalmopathy

**DOI:** 10.3390/ijms25063288

**Published:** 2024-03-14

**Authors:** Hsun-I Chiu, Shi-Bei Wu, Chieh-Chih Tsai

**Affiliations:** 1Department of Ophthalmology, Taipei Veterans General Hospital, Taipei 112, Taiwan; 2School of Medicine, National Yang Ming Chiao Tung University, Taipei 112, Taiwan; 3Biomedical Commercialization Center, Taipei Medical University, Taipei 110, Taiwan

**Keywords:** Graves’ orbital fibroblasts, Graves’ ophthalmopathy (GO), transforming growth factor-β1 (TGF-β1), oxidative stress, endoplasmic reticulum (ER), epigenetics

## Abstract

Graves’ ophthalmopathy (GO), or thyroid eye disease (TED), is the most frequent extrathyroidal manifestation of Graves’ disease (GD). Inflammation and subsequent aberrant tissue remodeling with fibrosis are important pathogenesis. There are many proposed mechanisms and molecular pathways contributing to tissue remodeling and fibrosis in GO, including adipogenesis, fibroblast proliferation and myofibroblasts differentiation, oxidative stress, endoplasmic reticulum (ER) stress, hyaluronan (HA) and glycosaminoglycans (GAGs) accumulation in the extracellular matrix (ECM) and new concepts of epigenetics modification, such as histone modification, DNA methylation, non-coding RNAs, and gut microbiome. This review summarizes the current understanding of ECM proteins and associated tissue remodeling in the pathogenesis and potential mediators for the treatment of GO.

## 1. Introduction

Graves’ ophthalmopathy (GO), also called thyroid eye disease (TED), is the most frequent extrathyroidal manifestation of Graves’ disease (GD) and is cosmetically disfiguring and potentially vision-threatening. Inflammation in the early phase, and subsequent aberrant tissue remodeling with fibrosis are important pathogeneses of GO [1]. Clinical manifestations, including lid retraction, proptosis, diplopia due to ocular motility, and compressive optic neuropathy restriction result from abnormal tissue remodeling of the orbital soft tissues. Severe abnormal orbital tissue remodeling and fibrosis may cause orbital deformity and vision loss (Figure 1). Current pharmacological agents are systemic steroids and immunomodulating agents, which may reduce inflammation but have limited effects on the long-term sequela [1]. Teprotumumab, a human monoclonal antibody inhibitor binding the extracellular alpha subunit of the IGF-1R, was approved by the U.S. Food and Drug Administration (FDA) in 2020 for patients with active moderate to severe GO [2]. However, the price of teprotumumab is extremely high and hearing loss is a major concern as an adverse event of teprotumumab [3]. Thyroidectomy could be considered for cases with recurrent hyperthyroidism; however, the evidence of thyroidectomy on GO is variable [4,5,6]. Thus, to identify novel targets of abnormal tissue remodeling and fibrosis in GO it is necessary to develop better therapeutics.

This review summarizes the current understanding of ECM proteins and associated tissue remodeling in the pathogenesis and potential mediators for the treatment of GO, and are summarized in Figure 2 and Table in Section 9.4, respectively.

## 2. Cell Mediators in GO

Activated GO fibroblasts are a hallmark in GO. During the active stage, the ligation of T cell receptor and major histocompatibility complex class II (MHC II) on B cells, and the interaction through CD40–CD154 (CD40 ligand) ligation between CD4+ T cells and fibroblasts, activates pro-inflammation and orbital fibroblasts [7]. CD4+ T lymphocytes in the orbit, and CD8+ T cells, macrophages, plasma cells and B cells in the extraocular muscle and adipose tissue produce lots of inflammatory mediators, such as interleukin (IL)-1β, IL-6, tumor necrosis factor (TNF)-α, transforming growth factor (TGF)-β and lipid mediators [8]. Additionally, autoantibodies against thyroid-stimulating hormone receptors (TSHR) and insulin-like growth factor-1 receptors (IGF-1 R) stimulate cAMP production and downstream mitogen-activated protein kinase/extracellular signal-regulated kinase (MAPK/ERK) pathways and phosphoinositide 3-kinase (PI3K) cascades [8,9], inducing orbital fibroblasts to secrete more inflammatory mediators, and increase the inflammatory cycle. Activated orbital fibroblasts differentiation into fat storing adipocytes through adipogenesis and myofibroblast transdifferentiation and ECM molecule secretion will be discussed in the following sections.

## 3. Adipogenesis and ECM of GO

Adipogenesis, the process of adipocyte (fat cell) development, results in the expansion of orbital adipose tissue and proptosis in GO. The differentiation of preadipocytes into mature adipocytes was associated with orbital fibroblasts. There are two distinct orbital fibroblasts: Thy-1(+) or Thy-1(−). Thy-1 is a classical T lymphocyte marker (CD90) expressed on fibroblasts. Fibroblasts expressing Thy-1 produce HA and inflammatory mediators, and are responsible for myofibroblast transdifferentiation. Thy-1(−) fibroblasts are capable of mature adipocyte differentiation. Lehmann et al. [10] demonstrated that the imbalance between Thy-1(+) and Thy-1(−) orbital fibroblasts facilitated adipogenesis in GO.

The PI3K pathway and lipid mediators induce the activation of peroxisome proliferator-activated receptor γ (PPARγ) and enhance adipogenesis. PPARγ is a nuclear receptor functioning as a transcription factor in lipid uptake and adipogenesis. Kumar et al. [11] demonstrated that several adipogenesis-associated mRNAs in orbital adipose tissues from GO patients, including leptin, adiponectin, PPARγ and preadipocyte factor-1, were elevated in parallel with TSH receptors. After IGF-1 stimulation, PPARγ was upregulated, and the PI3K pathway was activated [12]. Thus, the inhibitors of PPARγ may be a potential therapeutic target for GO. Notably, some type 2 diabetes medications, rosiglitazone or piogliatozone, which would activate PPARγ, may deteriorate GO [13].

## 4. Transforming Growth Factor-β1 (TGF-β1)-Induced Myofibroblast Transdifferentiation and ECM in GO

Multiple studies have established the involvement of TGF-β in tissue remodeling. TGF-β, along with insulin-like growth factor (IGF)-1, IL-4, and platelet-derived growth factor (PDGF), has been observed to increase the proliferation of orbital fibroblasts in GO compared to normal controls [14]. Among these mediators, TGF-β plays a crucial role in tissue remodeling. GO orbital fibroblasts show significantly higher levels of TGF-β protein and mRNA expression, and TGF-β is capable of triggering the transformation of Thy-1(CD90)-positive orbital fibroblasts into myofibroblasts [15]. There are three isoforms of TGF-β in mammals, with TGF-β1 being the primary contributor to tissue remodeling in GO [16].

TGF-β induces tissue fibrosis through either the canonical SMAD pathway or the non-canonical SMAD signaling pathway [17]. The TGF-β molecule consists of three parts: latent TGF-β binding protein (LTBP), latency-associated peptide (LAP), and activated TGF-β. Cleavage of LTBP and LAP by matrix metalloproteinases (MMP) releases activated TGF-β, which then interacts with TGF-β receptors. This interaction leads to the phosphorylation of SMAD 2/3, forming a complex with SMAD 4. The SMAD 2/3/4 complex enters the nucleus, initiating myofibroblast transdifferentiation and ECM production [18,19,20].

In the non-SMAD pathway, TGF-β activates non-Smad mediators, including the mitogen activated protein kinases (MAPK), phosphatidylinositol-3-kinase (PI3K) and Rho-like GTPases, to regulate gene expression [21]. The MAPK family consists of p38, c-Jun N-terminal kinase (JNK), and extracellular-signal-regulated kinase (ERK). In our prior study [22], the increased phosphorylation of p38 and JNK (but not ERK) after the induction of TGF-β1 was demonstrated, as well as the expression of connective tissue growth factor (CTGF), α-smooth muscle actin (α-SMA), fibronectin, tissue inhibitors of metalloproteinases 1 (TIMP-1) and TIMP-3. The levels of matrix metalloproteinases 2 (MMP-2) and MMP-9 were inhibited. Our study proved that the non-SMAD pathway of p38 and JNK contributed to the TGF-β-induced myofibroblast transdifferentiation in human GO Fibroblasts and ECM production. Inhibiting the p38 or JNK pathway may be a potential therapeutic strategy to treat fibrosis in GO.

When the activated Smad/non-Smad proteins enter the nucleus, they regulate transcription factors and increase ECM production, including collagen, CTGF, fibronectin, and alpha-smooth muscle actin (α-SMA). In our previous study, the expression of CTGF, fibronectin, and α-SMA in Graves’ orbital fibroblasts were stimulated after the induction of TGF-β1 [23,24,25]. The protein levels of fibronectin and α-SMA were increased after induction by recombinant human protein CTGF (rhCTGF) and decreased after knockdown of CTGF, which indicates CTGF is an essential downstream mediator for TGF-β1-induced myofibroblast transdifferentiation. Moreover, the elevated CTGF was correlated with clinical manifestations in GO patients [21].

Inhibiting TGF-β1 has potential as a therapeutic approach to target fibrosis; however, TGF-β is involved in not only tissue remodeling, but also cell proliferation and immune responses. The blockage of TGF-β1 might cause unexpected systemic side effects [26,27]. Thus, blockage downstream of TGF-β, such as the p38 or JNK pathway or CTGF, may bring a potential therapeutic target to treat fibrosis in GO.

## 5. Hyaluronan Formation by Orbital Fibroblasts

ECM consists of collagen and hyaluronan (HA). HA is the main contributor to volume expansion in GO because it occupies around 75,000 times the volume of that of an equivalent weight of collagen [28]. HA synthesis is by the actions of hyaluronan synthases (HAS). There are three isomers of HAS: HAS1, HAS2 and HAS3, and HAS2 mainly contributes to HA synthesis by orbital fibroblasts [29]. HA synthesis could be stimulated by inflammation. It has been reported that leukoregulin, IL-1, TNF-α, IFN-γ, TGF-β, IGF-1, PDGF and prostaglandins enhance HA synthesis in GO fibroblasts. Hyaluronic acids could induce MAPK and NF-κB phosphorylation; while HA-induced transcription of COX-2 was halted by the administration of MAPK or NF-κB inhibitors [30,31,32,33] and inhibited by methylprednisolone and dexamethasone. Additionally, HA synthesis could be stimulated by adipogenesis. Zhang et al. [34] showed that adipogenesis in orbital preadipocytes was associated with increased HAS2 transcripts and HA accumulation. Guo et al. [35] revealed the elevated expression of HAS mRNA in GO fibroblasts after treatment with mast cell derived prostaglandin D2 (PGD2) and PGJ2, which are both factors of adipogenesis. Although HA accumulation mainly leads to volume expansion in GO, the mechanism of HA degradation and the relationship between HA synthesis and degradation are not fully understood [34].

## 6. Oxidative Stress and Smoking in the ECM of GO

Oxidative stress represents an imbalance between the antioxidant defense system and reactive oxygen species’ (ROS) production, including superoxide anions, hydrogen peroxide, and hydroxyl radicals. When the production of ROS overwhelms the body’s antioxidant defenses, oxidative stress arises. Sources of ROS in GO include activated immune cells (such as infiltrating lymphocytes and macrophages) and orbital fibroblasts, which induce the expression of proinflammatory cytokines and chemokines [36]. Additionally, the GO autoantibodies could bind to orbital fibroblasts and activate NADPH oxidase, and further trigger the generation of ROS and exacerbate oxidative stress [37].

Although the exact pathogenic mechanisms of oxidative stress in GO are not fully specified, orbital fibroblasts are considered as the major effector cells. Our prior study found elevated urinary 8-hydroxy 2′-deoxyguanosine (8-OHdG) in active GO patients, which is an important biomarker of oxidative DNA damage [38]. Not only 8-OHdG, but also malondialdehyde (MDA), a product of lipid peroxidation, as well as the superoxide anions and hydrogen peroxide were increased in cultured orbital fibroblasts from patients with GO compared to controls, which indicated prominent oxidative DNA damage, lipid peroxidation, and ROS production in the pathogenesis of GO [39]. After the induction of hydrogen peroxide (H_2_O_2_), there were increased ROS contents (MDA, H_2_O_2_, and manganese-dependent superoxide dismutase) and imbalance of the ratio change between reduced (GSH) and oxidized glutathione (GSSG) in GO orbital fibroblasts [40]. Systemic glucocorticoids could decrease the level of urinary 8-OHdG in patients with active GO [35]. Similar results were obtained by Akarsu et al. [41], Abalovich et al. [42] and Bednarek et al. [43].

Cigarette smoking, an important risk factor for GO deterioration, has been established in multifactorial ways. Yuksel et al. [44] demonstrated the smoking effects on oxidative stress and mitochondrial homeostasis. Two proteins associated with proper mitochondrial function, paraoxonase (PON) and mitochondrial transcriptional factor A (MTFA), were found to be significantly decreased in GO smokers compared to GO nonsmokers or healthy subjects. Cawood et al. [45] showed cigarette induced hyaluronic acid production and adipogenesis via the synergizing effects of IL-1 and ROS. Additionally, smoking impairs endothelial function and disrupts the integrity of the vascular barrier within the orbit. Decreased superior ophthalmic venous blood flow velocity and choroidal blood circulation were found in GO [46,47]. The hypoxic environment might induce the production of hypoxia-inducible factor-1 (HIF-1) and stimulate the HIF-1-dependent adipogenesis pathway. The levels of HIF-1α were correlated with the clinical activity score (CAS) [48]. Similar results regarding smoking were demonstrated in our study. After induction by cigarette extracts, there was increased oxidative stress, fibrosis-related gene expression (apolipoprotein J, CTGF, and fibronectin), and intracellular levels of TGF-β1 and IL-1β in GO orbital fibroblasts [49].

## 7. Endoplasmic Reticulum (ER) Stress in the ECM of GO

The endoplasmic reticulum (ER) is an important intracellular organelle facilitating the conversion of nascent proteins to functional forms. Dysfunction of the ER leads to ER stress and triggers the unfolded protein response (UPR) of the chaperone protein through three main effector pathways, involving PKR-like ER kinase (PERK), activating transcription factor 6 (ATF6), and inositol-requiring enzyme 1α (IRE1α). Severe or prolonged ER stress has been proposed in multiple organs [50] with the development of fibrotic disorders, including liver, kidneys, heart, and lung, while ER stress in GO pathogenesis is in its infancy.

In orbital tissues from GO patients, Huang et al. revealed that ER stress-related gene (ATF6, PERK, and IRE1α) expression was higher than in control orbital tissues [51]. Further study showed that silencing PERK could reduce oxidative stress and adipogenesis in the GO orbital fibroblasts [52]. Our recent study (unpublished data) further demonstrates that ER protein TXNDC5 plays an important role in TGF-β1-induced myofibroblast trans-differentiation and ECM protein in GO orbital fibroblasts. We used lentivirus transfected TED orbital fibroblasts with small hairpin RNAs to knockdown TXNDC5 protein expression levels, and found that TXNDC5 knockdown attenuated TGFβ1-induced myofibroblast transdifferentiation and ECM protein production, whereas increasing TXNDC5 expression by recombinant TXNDC5 addition increased ECM protein expression.

## 8. Epigenetics and the Gut Microbiome in the ECM of GO

With advanced gene sequencing technology, the epigenetics in the pathogenesis of GO has been studied in recent years [53,54]. Although the research is still in its infancy, histone modification, DNA methylation and non-coding RNAs bring emerging insights and novel potential therapeutic strategies.

### 8.1. Histone Deacetylases (HDACs) and DNA Methylation

Histone deacetylases (HDACs) are enzymes that epigenetically control gene transcription through the modification of histone proteins. HDACs remove acetyl groups from histone lysine residues and reduce the expression of target genes. The downregulation of HDACs could induce the gene activity of proinflammation and immunity; thus, HDAC inhibitors (HDACis) may have therapeutic potential [55].

HDACs are known to be associated with malignancy, autoimmune diseases, as well as thyroid disease [53,56,57]. Limbach et al. [58] performed a genome-wide analysis in patients with Graves’ disease and showed dysregulated DNA methylation and histone modifications at T cell signaling genes. Sacristán-Gómez et al. [59] analyzed different kinds of HDAC and demonstrated that elevated HDAC9 and decreased Tip60 histone acetyltransferases (HAT) might suppress the activation of regulatory T cells and promote proinflammation.

Elevated DNA methylation in GO fibroblasts was demonstrated in recent years, after analysis of the proteomics and DNA methylation in orbital fibroblasts. Ekronarongchai et al. [60] revealed higher HDAC4 mRNA expression in GO orbital fibroblasts compared to those in normal subjects after stimulation with platelet-derived growth factor-BB (PDGF-BB). The expression of hyaluronan synthase 2 (HAS2), collagen type I alpha 1 chain (COL1A1), Ki67, α-SMA, and HA were reduced after silencing of HDAC4. Adding HDAC4i (tasquinimod) decreased the mRNA expression of HAS2 and α-SMA. Inhibitors of HDAC4 might be potential therapeutic targets for GO therapy.

### 8.2. MicroRNA in GO

Non-coding RNAs (ncRNA) are a class of RNA lacking protein-coding functions, and are classified into microRNA (miRNA), circular RNA (circRNA) and long non-coding RNA (lncRNA). Among these, MiR-146a is the most studied miRNA associated with GO. MiRNA is a small non-coding RNA that contains 21 to 23 nucleotides. It binds to 3′ untranslated regions of the target mRNA and post-transcriptionally regulates gene expression. MiR-146a was found to affect immune regulation, cell proliferation, differentiation, apoptosis and ECM [61,62]. Downregulated miR-146a expression would attenuate CD4+ T cell differentiation and proliferation and induce elevated IL-17 levels, which promotes orbital inflammation [63]. Jang et al. [62] demonstrated that miR-146a downregulates TGF-β-induced fibronectin, collagen Iα and α-SMA protein from GO orbital fibroblasts through Smad 4 and the tumor necrosis factor receptor-associated factor 6 (TRAF6) pathway. In addition to miR-146a, miR-155, causing the opposite effect against miR-146a, has been proposed to be associated with GO. Li et al. [64] revealed that increased miR-155 and decreased miR-146a promote orbital fibroblast proliferation. Woeller et al. [61] revealed that TSH stimulates proliferation of orbital fibroblasts through PI3K/Akt, miR-146a and miR-155. Recent findings suggest miR-155 upregulates collagen synthesis, which further induces the fibrosis process in GO [65]. With the high throughput of advanced technology, more non-coding RNA have been proposed in association with GO, such as miR-29, miR-21, MiR-27a, miR-27b, miR-130a [16,66,67,68]. Additional research is warranted to develop the crucial role of miRNA and to elucidate potential targets.

### 8.3. Other Non-Coding RNAs in GO

Circular RNA (CircRNA) is a closed loop RNA with joined of 3′ and 5′ ends. They acts as mRNA sponges and induce mRNA transcription [69]. Wu et al. [70] identified 163 circRNAs interacting with 607 mRNAs in GO orbital adipose, and circRNA_14936, circRNA_14940, and circRNA_12367 were associated with cytokine–cytokine interaction and ECM receptor interaction. Although the literature is limited, the co-expression of circRNA and mRNA was demonstrated in GO pathophysiology but warrants further evaluation.

Long non-coding RNAs (LncRNAs), involving more than 200 non-transcribed nucleotides, regulate gene expression in Hashimoto thyroiditis and other autoimmune diseases. Christensen et al. [71] firstly reported lncRNA (designated Heg) was negatively related to thyroid receptor autoantibodies in untreated Graves’ patients, which manifested anti-inflammatory effects. In the latest study, lncRNAs from the GO orbital tissue were identified as being associated with 52 kinds of ECM genes [72].

### 8.4. Gut Microbiome in GO

The relationship between gut microbiota and autoimmune diseases, including Graves’ disease (GD) or GO, is an area of growing interest and research. Literature revealed alterations in the gut microbiome that may induce GO by impacting the levels of thyroid-stimulating hormone receptor antibodies (TSH Ab) and fostering an imbalance in Th17/Treg [53,73,74]. Experimental findings revealed heterogeneity in the gut microbiome in a TSHR-immunized GD/GO mouse model and a reduction in disease severity with oral antibiotic vancomycin treatment [75].

However, fecal microbiota transplantation from GD/GO patients exacerbated the condition [73,75]. Studies revealed GD/GO patients exhibit distinct differences in microbiota composition compared to healthy controls and the composition of microbiota could be impacted by various factors, such as glucocorticoids, antithyroid drugs and immunosuppressants [75]. Significant differences in gut microbiota composition caused by various factors could explain variations in study results and clinical application. To elucidate the link between GO and the gut microbiome warrants further investigation.

In summary, histone modification, DNA methylation, and non-coding RNAs may play a crucial role in GO, but further evaluation is warranted to elucidate the applications.

## 9. Treatment for Tissue Remodeling in GO

### 9.1. Biologic Agents

Current pharmacological agents are systemic steroids and immunomodulating agents, which may reduce inflammation but have limited effects on the long-term tissue remodeling. ECM production, tissue remodeling and fibrosis of the orbital soft tissues are responsible for the clinical manifestations of proptosis, lid retraction and compressive optic neuropathy in GO. They highlight the urgency required to develop novel targets against GO related fibrogenesis.

Teprotumumab is a monoclonal antibody that targets the insulin-like growth factor-1 receptor (IGF-1R). It decreases levels of the receptors of TSH and IGF-1 on orbital fibrocytes and attenuates TSH-stimulated pro-inflammatory cytokines, including IL-6, IL-8, and TNF-α. A randomized, double-blind, placebo-controlled, multicentered, phase III trial enrolled 41 patients with active, moderate-to-severe GO [2]. Either intravenous teprotumumab or placebo infusions were prescribed every 3 weeks for a total of eight infusions. After 24 weeks, patients receiving teprotumumab had significant improvement of proptosis (83% vs. 10%, *p* < 0.001). This “breakthrough therapy” teprotumumab was approved by the US FDA in 2020. However, the high price is difficult to afford, and the infusion requirement limits the extensibility. The side effects of hyperglycemia and hearing impairment need to be solved [76]. Exploring other potential therapeutic targets is necessary.

Other biologic agents, including rituximab and tocilizumab, have been reported. Rituximab is a monoclonal antibody against CD20 targeting B cell lymphocytes. It works by depleting B cells, which leads to the reduction in autoantibodies and pro-inflammatory cytokines. Rituximab has been used clinically, while the results from the randomized control trials (RCT) are inconsistent. Salvi et al. [77] demonstrated better outcome of ocular motility and quality of life in cases of moderate to severe GO after use of intravenous rituximab compared to intravenous methylprednisolone. However, Stan et al. [78] showed no benefit in cases of moderate to severe GO after use of intravenous rituximab compared to placebo. Tocilizumab is a humanized antibody against IL-6 in T cell and B cell activation. Additionally, IL-6 may change the ECM remodeling by inducing the expression of the thyrotropin receptor and thyroid-stimulating immunoglobulin (TSI) in orbital fibroblasts, leading to the differentiation into myofibroblasts or adipocytes [79]. By blocking IL-6, tocilizumab decreases B cell activation, T cell recruitment and orbital fibroblast activation. However, there are no published randomized clinical trials. Biologic agents show promise in treating GO, the benefits and side effects warrant evaluation of more RCTs.

### 9.2. Potential Therapeutic Target—Antioxidants

Multiple potential antioxidants have been described in GO, including selenium, pentoxifylline, quercetin, enalapril, allopurinol, nicotinamide, vitamin C, N-acetylcysteine, melatonin, β-carotene, and statins [80,81,82]. A recent systematic review [80] included four clinical and ten in vitro studies from 1993 to 2018, and demonstrated that selenium was the only antioxidant based on in vitro and randomized controlled trial studies. The following paragraphs highlight the potential therapeutic effects of selenium, pentoxifylline, allopurinol, and nicotinamide, because they were reported to be effective in both clinical and experimental reports. β-carotene, retinol, vitamin E, vitamin C, melatonin, resveratrol, N-acetyl-l-cysteine, and quercetin showed potential in in vitro studies, but need more clinical data.

#### 9.2.1. Selenium (Se)

Selenium is a trace mineral that is incorporated into several selenoproteins, such as glutathione peroxidase, thioredoxin reductase and iodothyronine deiodinases (D1, D2 or D3). Selenoproteins have antioxidative and immune-modulating effects through activating T cells and reducing the release of tumor necrosis factor α (TNFα) and cyclooxygenase 2 (COX2) [83,84]. Thus, Se as a supplementary modality, may have a role in GO treatment.

One randomized control trial (RCT) [85] founded by the European Group on Graves’ Orbitopathy (EUGOGO) compared the effect of selenium (100 μg twice daily), pentoxifylline (600 mg twice daily) or placebo in 159 patients with mild GO. The trial demonstrated noteworthy enhancements in both quality of life and ocular involvement, coupled with a reduced progression rate in patients with mild GO following a 6-month selenium supplementation. However, this recommendation was not endorsed by the American Thyroid Association (ATA) and the European Thyroid Association (ETA), since selenium deficiency is not present in the US and Europe [86,87]. Additionally, Se could be consumed by daily intake, such as egg, milk and meat. The risk between long-term Se intake and diabetes was uncertain [88,89] and the role of Se on moderate to severe GO found insufficient evidence [90,91]. More RCT was needed to confirm the efficacy and safety.

Despite the uncertainty of Se supplementation, its ability to maintain adequate Se serum levels and to avoid severe Se deficiency deserves clinical attention.

#### 9.2.2. Pentoxifylline

Pentoxifylline is a methyl-xanthine derivative and is approved by the FDA to improve ischemic symptoms from peripheral arterial diseases. In 1993, Chang et al. [92] demonstrated that pentoxifylline inhibited the proliferation of fibroblasts from GO patients, as well as glycosaminoglycan synthesis in vitro. Further studies revealed that cytokine induced HLA-DR expression, serum levels of GAG, TNF-α, anti-TSH-receptor, anti-eye muscle, anti-thyroglobulin and anti-thyroid peroxidase antibodies were suppressed by pentoxifylline [93,94]. A prospective RCT [95], held in 2004, with a total of 18 inactive GO patients, distributed patients to two groups randomly: one group with pentoxifylline 1200 mg orally/day for 6 months, and the other group with placebo for initial 6 months sequentially pentoxifylline for another 6 months. The study demonstrated a significant change in proptosis after 3 months and 6 months of pentoxifylline use. However, inconsistent results were found in the large-sampled double blinded RCT [85] from the EUGOGO group, which revealed the insignificant effect of pentoxifylline (600 mg twice daily for 6 months) in mild GO patients. More RCT were needed to confirm the efficacy in different severity GO patients (active, inactive; mild, moderate and severe).

#### 9.2.3. Nicotinamide and Allopurinol

Nicotinamide, an amide form of vitamin B3, is a component of the coenzymes of nicotinamide adenine dinucleotide (NAD+), nicotinamide adenine dinucleotide (NADH), nicotinamide adenine dinucleotide phosphate (NADP+), and nicotinamide adenine dinucleotide phosphate (NADPH), in the oxidation–reduction assay. It was found that nicotinamide could inhibit HLA-DR expression on orbital fibroblasts from GO patients and suppressed superoxide-induced fibroblast proliferation. It also stimulated cytokine-induced activation and enhanced apoptosis through the Fas ligand on fibroblasts in vitro [96,97].

Allopurinol is an inhibitor of xanthine oxidase. Uric acid is a powerful scavenger of ROS; through reducing purine metabolism, allopurinol is regarded as an antioxidant. Experimental studies revealed allopurinol reduced the level of xanthine oxidase, malondialdehyde, glutathione and nitric oxide derivatives in hyperthyroidism in rats [98,99].

In a non-randomized prospective study [100], oral allopurinol (300 mg daily) and nicotinamide (300 mg daily) were used in 11 GO patients. After 3 months of usage, significant improvements in visual acuity (82%) and in the soft tissue inflammation (90%) were found, compared to the placebo group. However, limitations were the small numbers, newly diagnosed GO patients from mild to severe severity and two antioxidants used simultaneously. Large-numbered RCT with separate therapy is warranted to evaluate the efficacy. Additionally, allopurinol-induced subclinical hypothyroidism (odds ratio 1.51) was found in a recent large-scaled observational study [101]. Because allopurinol may induce the elevation of thyroid-stimulating hormone levels, the safety of long-term use warrants more clinical attention.

### 9.3. Potential Therapeutic Target—Pirfenidone

Pirfenidone is indicated for the treatment of idiopathic pulmonary fibrosis. Regarding GO, it was found that pirfenidone attenuates levels of cyclooxygenase 2, prostaglandin E2 and tissue inhibitors of metalloproteinase (TIMP)-1 in IL-1β–induced orbital fibroblasts [102,103,104]. In our previous study [105], we found pretreatment of pirfenidone decreased the levels of α-SMA, CTGF, fibronectin, and collagen type I on TGF-β1-induced orbital fibroblasts, through the p38 and c-Jun N-terminal kinase (JNK) pathways. The role of pirfenidone on ECM homeostasis and tissue remodeling provides a potential strategy.

### 9.4. Other Potential Targets

Factors influencing endoplasmic reticulum stress and oxidative stress, and the influence of smoking have been recognized as critical components in the disease’s development and severity. Epigenetic regulators, particularly histone deacetylases (HDACs), have emerged as key players in modulating the immune responses and fibrotic processes that drive GO. In addition, Chen et al. [106] demonstrated that targeting CD40–CD40L signaling by a specific RNA aptamers (CD40Apt) could reduce the levels of CD40, collagen I, TGF-β, and α-SMA in orbital muscle and adipose tissues of model mice. This represents a promising antagonist of CD40–CD40L signaling for TAO treatment. Further, the emergence of advanced gene sequencing technologies has illuminated the roles of microRNAs (miRNAs), circular RNAs (circRNAs), and long non-coding RNAs (lncRNAs) in the pathogenesis of GO, offering new avenues for understanding and potentially controlling the disease. We summarize the potential therapies in Table 1.

## 10. Concluding Remarks

In this review, we have demonstrated a picture of the crucial roles and emerging issues related to ECM proteins in GO pathogenesis. To understand the mechanisms of ECM protein dysregulation in GO, it is important to elucidate the molecular basis of the excessive proliferation and ECM production by GO orbital fibroblasts and the involvement of certain signaling pathways, leading to tissue expansion/remodeling and fibrosis in GO.

Current pharmacological agents are systemic steroids and immunomodulating agents, which may reduce inflammation but have limited effects on the long-term sequela. While teprotumumab, a monoclonal antibody targeting the insulin-like growth factor-1 receptor (IGF-1R), has shown remarkable success in managing active proptosis in GO, its cost, adverse effects, and infusion requirements emphasize the need for additional therapeutic options. Multiple antioxidants and antifibrotic agents, such as pirfenidone, hold promise as potential therapies or prophylactic agents, while warranting long-term and large-scaled clinical trials. The identification of novel biomarkers, including antagonists in ER stress, the PPARγ pathway, small molecules of histone and non-coding RNA, provide a more comprehensive understanding of the disease pathways. The ideal scenario would involve the oral or subcutaneous delivery of these novel therapies with minimal side effects. Broadening the concepts of the ECM and tissue fibrosis will help in the development of new therapeutic targets in GO.

## Figures and Tables

**Figure 1 ijms-25-03288-f001:**
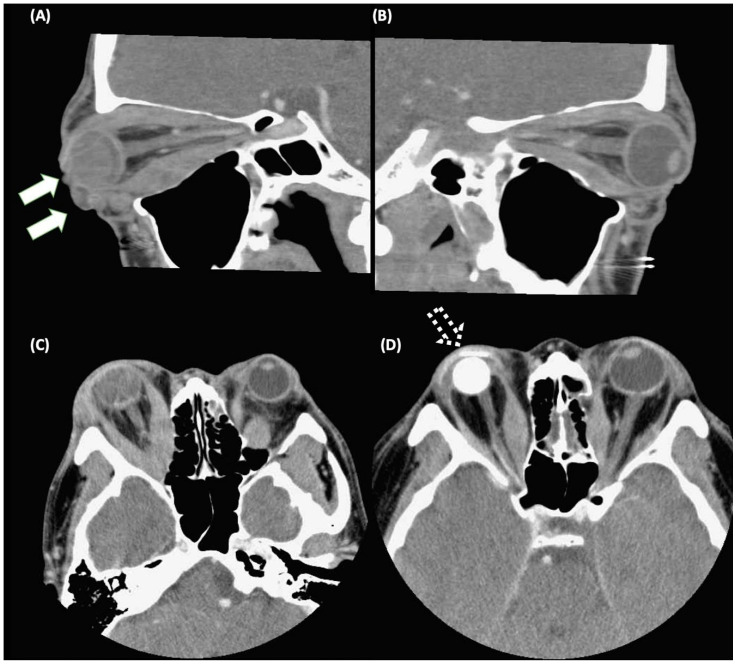
Computed tomography images in a patient with Graves’ ophthalmopathy (GO). (**A**) Exposure keratopathy with eyeball rupture (solid arrows) due to severe extraocular muscle hypertrophy and adipogenesis in the right eye. (**B**) Extraocular muscle hypertrophy in the left eye. (**C**,**D**) The patient received an evisceration with an implant (as dashed arrows) of the right eye, and orbital decompression of the left eye. The pre-operative image and post-operative image were demonstrated in (**C**) and (**D**), respectively.

**Figure 2 ijms-25-03288-f002:**
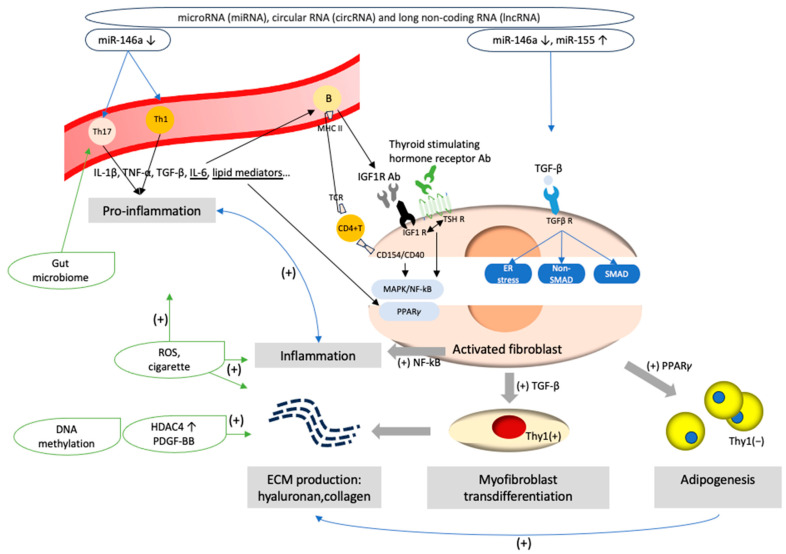
Pathogenesis of Graves’ ophthalmopathy. (1) Aberrant epigenetic modifications, such as dysregulation of microRNA (miRNA)-146, miRNA 155, circular RNAs and long non-coding RNA, might trigger pro-inflammatory cascades and disturb the expression of signaling molecules critical for myofibroblast transdifferentiation and adipogenesis processes in orbital fibroblasts. (2) The interaction through CD40–CD154 ligation and T cell receptor (TCR)–major histocompatibility complex class II (MHC II) ligation activates pro-inflammation and orbital fibroblasts with the secretion of inflammatory mediators, such as IL-1β, IL-6, tumor necrosis factor (TNF)-α, transforming growth factor (TGF)-β, and lipid mediators. Orbital fibroblasts further upregulate the release of pro-inflammatory cytokines. IL-6 modulates B cell immunoglobulin secretion. Autoantibodies of insulin-like growth factor-1 (IGF-1) and thyroid-stimulating hormone receptor antibodies (TSH Ab) interact with their receptors and activate the thyrotropin/IGF-1 receptor complex, promoting the further mitogen-activated protein kinase (MAPK) and nuclear factor kappa-light-chain-enhancer of activated B cells (NF-κB) downstream signaling pathways, which induce orbital fibroblast proliferation and inflammation. (3) The interaction of thyrotropin/IGF-1 receptor complex and lipid mediators upregulates peroxisome proliferator-activated receptor-γ (PPAR-γ) expression, and induce adipocyte differentiation from Thy1 negative orbital fibroblasts. (4) TGF-β binds to its receptors and activates Smad/non-Smad transcription factors. Thy1 positive orbital fibroblasts are activated and result in myofibroblast transdifferentiation. (5) Contents of the extracellular matrix, such as hyaluronan (HA) or collagen, are produced by activated orbital fibroblasts. Aberrant histone modification and DNA methylation, such as the stimulation of platelet-derived growth factor-BB (PDGF-BB) and histone deacetylases (HDAC) 4, would promote the mRNA expression of HA production and pro-inflammation. Factors of adipogenesis, such as prostaglandin D2 (PGD2) and PGJ2, enhance ECM production. (6) Cigarette enhances oxidative stress and upregulate TGF-β1, IL-1β, adipogenesis and the fibrosis-related gene expression in orbital fibroblasts. (7) Endoplasmic reticulum (ER) stress promotes fibrogenesis after the stimulation of TGF-β. (8) The gut microbiome may foster an imbalance in Th17 and T regulatory cells and impact the levels of TSH Ab.

**Table 1 ijms-25-03288-t001:** Current and potential treatment for Graves’ ophthalmopathy (GO).

**Therapy, FDA Approved ** **[Reference]**	**Mechanism**	**Study Design**	**Disease Status**	**Outcome**	**Adverse Effect**
Teprotumumab [2]	Monoclonal antibody against insulin-like growth factor-1 receptor (IGF-1R)	RCT (*n* = 41) iv 10 mg/kg initially, followed by 20 mg/kg in 3 weeks for 7 additional infusions	Moderate to severe	Improved QOL and proptosis Improvement (83% vs. 10% *p* < 0.001) after 24 weeks	(1) Hyperlipidemia (2) Hearing impairment
**Therapeutic agents with RCT [reference]**	**Mechanism**	**Study design**	**Disease Status**	**Outcome**	**Adverse effect**
Rituximab [77]	Monoclonal antibody against CD20	RCT (*n* = 31) iv 2000 mg or 500 mg control: iv methylprednisolone	Moderate to severe	Better ocular motility and QOL after 24 weeks	(1) Infusion reaction (2) Transient hypotension
[78]		RCT (*n* = 13) iv 1000 mg twice in 2 weeks control: placebo	Moderate to severe	No benefit	Infectious bronchitis, conjunctivitis, vasculitis, optic neuropathy, gastrointestinal disorder
Selenium [85]	Antioxidant	RCT (*n* = 54) 200 μg/day for 6 months	Mild	Improved QOL and ocular sign, decreased progression	(−)
Pentoxifylline [85]	Antioxidant	RCT (*n* = 48) 1200 mg/day for 6 months	Mild	No benefit	Mild gastrointestinal and skin disorders
[95]		RCT (*n* = 9) 1200 mg/day for 6 months	Inactive	QOL and proptosis improvement	Mild gastrointestinal disorder
Nicotinamide and Allopurinol [100]	Antioxidant	Nonrandomized comparative study (*n* = 11) Oral allopurinol (300 mg/day) and nicotinamide (300 mg/day) for 3 months	Mild to moderate	Improved soft tissue inflammation	(−)
**Potential therapeutic agents without RCT ** **[reference]**	**Mechanism**				
Tasquinimod [60]	Inhibitor of histone deacetylases 4 (HDAC4), which decreased the mRNA expression of hyaluronan synthase				
Tocilizumab [79]	Monoclonal interleukin-6 receptor antagonist				
Pirfenidone [102,103,104,105]	(1) Decreased inflammation by attenuation of COX-2, prostaglandin E2 (2) Decreased ECM production				
CD40Apt [106]	Specific binding affinity to CD40 represents a promising inhibitor of the CD40-CD40L signaling				
**Potential therapeutic targets [reference]**	**Mechanism**				
Antagonist of peroxisome proliferator-activated receptor *γ* (PPAR*γ*) [10,11,12]	Inhibition of adipogenesis				
Antagonist of transforming growth factor-β1 (TGF-β1) downstream [22,23,24,25]	Inhibition of TGF-β-Induced myofibroblast transdifferentiation in orbital fibroblasts and ECM production				
Antagonist of thioredoxin domain-containing 5 (TXNDC5) [our study, not published]	Reduction in ER stress and tissue remodeling				
Non-coding RNA [61,62,63,64,65,66,67,68,69,70,71,72]	Decreased levels of mi146a: decreased inflammation, decreased fibrosis in GO Increased levels of mi155: decreased fibrosis in GO others (miR-29, miR-21, MiR-27a, miR-27b, miR-130a, circRNA, LncRNA)				

FDA: Food and Drug Administration; RCT: randomized control trial; iv: intravenous; QOL: quality of life; ECM: extracellular matrix; COX-2: cyclooxygenase 2; miRNA: microRNA; circRNA: circular RNA; lncRNA: long non-coding RNA.

## Data Availability

Not applicable.

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
