# Peer review of "The Role of Fibrogenesis and Extracellular Matrix Proteins in the Pathogenesis of Graves’ Ophthalmopathy"

_ijms, 2024, doi:10.3390/ijms25063288_

Round 1
Reviewer 1 Report
Comments and Suggestions for Authors
Reviewer comments (ijms-2872935)
The manuscript provides interesting findings concerning the role of fibrogenesis and extracellular matrix proteins in pathogenesis of Graves' ophthalmopathy. The authors achieved well the results clearly demonstrated that therapeutic efficacy and pathophysiological modulators of other risk factors in GO. Therefore, I recommend the publication of this manuscript after a minor revision would be done.
Comments are presented below:
- Abstract lines 12-19 repeated again in introduction lines 39-45, need to be deleted in introduction part, as it's meaningless.
- Instead of lines-39-45, page-10, lines-391-412 should be moved to this introduction part, since they provide appropriate information. However, figure 2 and its captions should be placed under summary or concluding remarks (subheading 10).
- The references in the text part, which do not meet the journal's criteria, also need to be revised.
- Lines-76, 80-83, etc...unnecessary the text mentioning in italics, next all those short form proteins, receptors and genes need to give their full form when first introduced in the texts, for example, TGF-b, a-SMA, SMAD, PI3K, PPARg, IGF-1, TXNDC5, etc. Then, in the next chapter, they must be followed up in their short form.
- In lines 86-128, TGF-b explanations are repeatedly mentioned, and the sentences need to be rewritten and simplified. Particularly lines 95 and 97...
- Lines 198-200 the authors explained with TXNDC5, its need to elaborately discuss the GO prevention role and its mechanism.
- Lines-219-223 DNA methylation sentences have to move into the introduction section. Line-228 shows in the completed sentence, it needs to explain the HDAC4 inhibitor therapeutic target for what?
- In the summary or concluding remarks, lines 265-266 and 381-388 must be moved under subheading 10.
- It needs to be simplified and reformatted with a single heading and table format. It also needs to have more references under each title.
- It is necessary to reformat references according to the instructions provided by the IJMS journal.
Author Response
Response to Reviewer 1 Comments
Thank you for taking the time to review our manuscript and offer valuable feedback and comments. On behalf of all the authors of this manuscript (manuscript number ijms-2872935), I submit our responses to the reviewers’ comments and suggestions. The revised parts are highlighted in yellow for your convenience.
Point-by-point response to Comments and Suggestions for Authors
Comment 1: Abstract lines 12-19 repeated again in introduction lines 39-45, need to be deleted in introduction part, as it's meaningless.
Response 1: Thank you for your suggestion. Based on the comments from 3 reviewers, we revised the introduction part as follows (page 1 of 15, line 42): Thus, to identify novel targets of abnormal tissue remodeling and fibrosis in GO is necessary to develop better therapeutics. This review summarizes the current understanding of extracellular matrix proteins and associated tissue remodeling in the pathogenesis and potential mediators for the treatment of GO, and summarized in the figure 2 and table 1, respectively.
Comment 2: Instead of lines-39-45, page-10, lines-391-412 should be moved to this introduction part, since they provide appropriate information. However, figure 2 and its captions should be placed under summary or concluding remarks (subheading 10).
Response 2: Thank you for highlighting this. We revised the introduction part as follows (page 1 of 15, line 42): Thus, to identify novel targets of abnormal tissue remodeling and fibrosis in GO is necessary to develop better therapeutics. This review summarizes the current understanding of extracellular matrix proteins and associated tissue remodeling in the pathogenesis and potential mediators for the treatment of GO, and summarized in the figure 2 and table 1, respectively.
We also revised the concluding remarks as follows (page 10 of 15, line 415): In this review, we have demonstrated a picture of the crucial role and emerging issues related to extracellular matrix proteins in GO pathogenesis. To understand the mechanisms of extracellular matrix proteins dysregulation in GO, it is important to elucidate the molecular basis of the excessive proliferation and extracellular matrix production by GO orbital fibroblasts and the involvement of certain signaling pathways, leading to tissue expansion/remodeling and fibrosis in GO.
Current pharmacological agents are systemic steroids and immunomodulating agents, which may reduce inflammation but have limited effects on the long-term sequela. While teprotumumab, a monoclonal antibody targeting the insulin-like growth factor-1 receptor (IGF-1R), has shown remarkable success in managing active proptosis in GO, its cost, adverse effects, and infusion requirements emphasize the need for additional therapeutic options. Multiple antioxidants and antifibrotic agents, such as Pirfenidone, hold promise as potential therapies or prophylactic agents, while warranting long-term and large-scaled clinical trials. The identification of novel biomarkers, including antagonists in ER stress, PPARγ pathway, small molecules of histone and non-coding RNA, provide a more comprehensive understanding of the disease pathways. The ideal scenario would involve the oral or subcutaneous delivery of these novel therapies with minimal side effects. Broadening the concepts of the ECM and tissue fibrosis will help in the development of new therapeutic targets in GO.
Comment 3: The references in the text part, which do not meet the journal's criteria, also need to be revised.
Response 3: We have revised the references format according to the instructions provided by the IJMS journal. Thank you for your observation.
Comment 4: Lines-76, 80-83, etc...unnecessary the text mentioning in italics, next all those short form proteins, receptors and genes need to give their full form when first introduced in the texts, for example, TGF-b, a-SMA, SMAD, PI3K, PPARg, IGF-1, TXNDC5, etc. Then, in the next chapter, they must be followed up in their short form.
Response 4: Thank you for your observation. We have revised the typing error, the italics and the short form as follows (page 3 of 15, line 77): The PI3K pathway and lipid mediators induce the activation of peroxisome proliferator-activated receptor γ (PPARγ) and enhance the adipogenesis. PPARγ is a nuclear receptor functioning as a transcription factor in lipid uptake and adipogenesis. Kumar et al demonstrated several adipogenesis-associated mRNA in orbital adipose tissues from GO patients, including leptin, adiponectin, PPARγ and preadipocyte factor-1, were elevated in parallel with TSH receptors. After IGF-1 stimulation, PPARγ was up-regulated, and PI3K pathway was activated. Thus, the inhibitors of PPARγ may bring a potential therapeutic target for GO. Notably, some type 2 diabetes medication, rosiglitazone or piogliatozone, which would activate PPARγ, might deteriorate GO.
Comment 5: In lines 86-128, TGF-b explanations are repeatedly mentioned, and the sentences need to be rewritten and simplified. Particularly lines 95 and 97…
Response 5: The sentences have been rewritten and simplified as follows (page 3 of 15, line 88): Multiple studies have established the involvement of TGF-β in tissue remodeling. TGF-β, along with insulin-like growth factor (IGF)-1, IL-4, and platelet-derived growth factor (PDGF), has been observed to increase the proliferation of orbital fibroblasts in Graves' ophthalmopathy (GO) compared to normal controls. Among these mediators, TGF-β plays a crucial role in tissue remodeling. GO orbital fibroblasts show significantly higher levels of TGF-β protein and mRNA expression, and TGF-β is capable of triggering the transformation of Thy-1(CD90)-positive orbital fibroblasts into myofibroblasts. There are three isoforms of TGF-β in mammals, with TGF-β1 being the primary contributor to tissue remodeling in GO.
TGF-β induces tissue fibrosis through either the canonical SMAD pathway or the non-canonical SMAD signaling pathway. The TGF-β molecule consists of three parts: latent TGF-β binding protein (LTBP), latency-associated peptide (LAP), and activated TGF-β. Cleavage of LTBP and LAP by matrix metalloproteinases (MMP) releases activated TGF-β, which then interacts with TGF-β receptors. This interaction leads to the phosphorylation of SMAD 2/3, forming a complex with SMAD 4. The SMAD 2/3/4 complex enters the nucleus, initiating myofibroblast transdifferentiation and ECM production.
Comment 6: Lines 198-200 the authors explained with TXNDC5, its need to elaborately discuss the GO prevention role and its mechanism.
Response 6: We discuss it elaborately and revised the paragraphs as follows (page 5 of 15, line 200): Our recent study (unpublished data) further demonstrates that ER protein TXNDC5 play an important role in TGF-β1-induced myofibroblast trans-differentiation and ECM protein in GO orbital fibroblasts. We lentivirus transfected TED orbital fibroblasts with small hairpin RNA to knockdown TXNDC5 protein expression levels, and found that TXNDC5 knockdown of attenuated TGFβ1-induced myofibroblast trans-differentiation and ECM protein upregulation, whereas increasing TXNDC5 expression by recombinant TXNDC5 addition increased ECM protein expression.
Comment 7: Lines-219-223 DNA methylation sentences have to move into the introduction section. Line-228 shows in the completed sentence, it needs to explain the HDAC4 inhibitor therapeutic target for what?
Response 7: We revised the manuscript as follows:
(page 1 of 15, line 17): new concepts of epigenetics modification, such as histone modification, DNA methylation, non-coding RNAs, and gut microbiome.
(page 5 of 15, line 233): The inhibitor of HDAC4 might be a potential therapeutic target for GO therapy.
Comment 8: In the summary or concluding remarks, lines 265-266 and 381-388 must be moved under subheading 10.
Response 8: We revised lines 265-266, 381-388 and revised the abstract as follows (page 1 of 15, line 17): new concepts of epigenetics modification, such as histone modification, DNA methylation, non-coding RNAs, and gut microbiome.
Comment 9: It needs to be simplified and reformatted with a single heading and table format. It also needs to have more references under each title.
Response 9: We have revised table 1. Thank you for your observation.
Comment 10: It is necessary to reformat references according to the instructions provided by the IJMS journal.
Response 10: We have revised the references format. Thank you for your observation.
Response to Comments on the Quality of English Language
Point: English language fine. No issues detected
Response: Thank you for taking the time to review our manuscript.

Reviewer 2 Report
Comments and Suggestions for Authors
This review is a good summary of a disease that is rare and not overly evaluated within this field. The inclusion of an epigenetic evaluation of the disease was interesting.
A lot of the therapeutics evalauted were mAbs or drugs. Are there any cell therapies/ biomaterial therapies that are currently being evaluated for treatment?
How do these therapies compare to let's say a thyroidectomy?
There has been a lot of talk recently on the gut microbiome and how that has affected a bunch of diseases. How do you think these two might have a link?
Author Response
Response to Reviewer 2 Comments
Thank you for taking the time to review our manuscript and offer valuable feedback and comments. On behalf of all the authors of this manuscript (manuscript number ijms-2872935), I submit our responses to the reviewers’ comments and suggestions. The revised parts are highlighted in yellow for your convenience.
Point-by-point response to Comments and Suggestions for Authors
Comment 1: A lot of the therapeutics evalauted were mAbs or drugs. Are there any cell therapies/ biomaterial therapies that are currently being evaluated for treatment?
Response 1: Thank you for highlighting the point. The field of regenerative medicine has been advancing. A monoclonal antibody targeting the neonatal Fc receptor (FcRn), and thyroid-stimulating hormone receptor (TSHR) antagonists have been proposed for the treatment of thyroid eye disease (TED). FcRn plays a crucial role in transporting immunoglobulin G (IgG) and preventing its lysosomal degradation. TSHR antagonists selectively inhibit TSH-stimulated signaling [reference 1]. The monoclonal antibody IMVT-1401 disrupts the interaction between IgG and FcRn, enhancing the catabolism of IgG and potentially increasing the degradation of pathogenic autoantibodies against thyroid-stimulating hormone receptor (TSHR) and insulin-like growth factor-1 receptor (IGF-1R). A phase IIa clinical trial (ASCEND-GO 1, NCT03922321, IMVT-1401, an FcRn antibody) assessed the safety and improvement in seven adult patients with moderate-to-severe, active TED. However, the trial was halted due to observed elevated total cholesterol and low-density lipoprotein [reference 2]. As our latest update of cell therapies and biomaterial therapies, there have been advancements but without clinical applications yet.
reference 1: Neumann S, Nir EA, Eliseeva E, et al. A selective TSH receptor antagonist inhibits stimulation of thyroid function in female mice. Endocrinology 2014; 155: 310–314.
reference 2: Wang Y, Tian Z, Thirumalai D, et al. Neonatal Fc receptor (FcRn): a novel target for therapeutic antibodies and antibody engineering. J Drug Target 2014; 22: 269–278.
Comment 2: How do these therapies compare to let's say a thyroidectomy?
Response 2: Thank you for highlighting. A meta-analysis comprising 7241 patients in active TED showed a 100% cure rate among patients who underwent total thyroidectomy and a 60% euthyroid state and an 8% rate of persistent or recurrent hyperthyroidism among patients who underwent near total thyroidectomy[ref 1]. However, the natural history of GO is variable with a course often independent of the clinical course of thyrotoxicosis [ref 2]. Abrupt alteration in thyroid functional status can be complicated with exacerbation of Graves’ ophthalmopathy (GO) [ref 3] and hypothyroidism is especially associated with exacerbation of ophthalmopathy [ref 4]. Thus, the importance is to maintain euthyroid state in the active phase. Thyroidectomy could be considered for cases with hyperthyroidism intolerance to antithyroid drugs, with a nodule with abnormal cytology and etc, while the effect of thyroidectomy on Graves’ exophthalmos is variable [references 5,6,7]. We have revised the manuscript as follows (page 1 of 15, line 38): Thyroidectomy could be considered for cases with recurrent hyperthyroidism; however, the evidence of thyroidectomy on GO is variable.
reference 1: Palit TK, Miller CC 3rd, Miltenburg DM. The efficacy of thyroidectomy for Graves' disease: A meta-analysis. J Surg Res. 2000;90(2):161-165. doi:10.1006/jsre.2000.5875
reference 2: Burch HB, Wartofsky L. Graves' ophthalmopathy: current concepts regarding pathogenesis and management. Endocr Rev. 1993;14(6):747-793. doi:10.1210/edrv-14-6-747
reference 3: Vasanthakumar P, Kumar P, Rao M. Anthropometric analysis of palpebral fissure dimensions and its position in South Indian ethnic adults. Oman Med J. 2013;28:26–32. doi: 10.5001/omj.2013.06.
reference 4: Stan MN, Durski JM, Brito JP, Bhagra S, Thapa P, Bahn RS. Cohort study on radioactive iodine-induced hypothyroidism: implications for Graves’ ophthalmopathy and optimal timing for thyroid hormone assessment. Thyroid. 2013;23:620–625. doi: 10.1089/thy.2012.0258.
reference 5: RYAN H. Thyroidectomy and thyrotropic exophthalmos (exophthalmic ophthalmoplegia) a review of 1001 thyroidectomies. Br J Ophthalmol. 1949;33(12):769-773. doi:10.1136/bjo.33.12.769
reference 6: Bhargav PRK, Sabaretnam M, Kumar SC, Zwalitha S, Devi NV. Regression of Ophthalmopathic Exophthalmos in Graves' Disease After Total Thyroidectomy: a Prospective Study of a Surgical Series. Indian J Surg. 2017;79(6):521-526. doi:10.1007/s12262-016-1516-8
reference 7: Liu ZW, Masterson L, Fish B, Jani P, Chatterjee K. Thyroid surgery for Graves' disease and Graves' ophthalmopathy. Cochrane Database Syst Rev. 2015;(11):CD010576. Published 2015 Nov 25. doi:10.1002/14651858.CD010576.pub2
Comment 3: There has been a lot of talk recently on the gut microbiome and how that has affected a bunch of diseases. How do you think these two might have a link?
Response 3: This is an interesting field. The relationship between gut microbiota and autoimmune diseases, including Graves’ disease/Graves’ ophthalmopathy (GD/GO), is an area of growing interest and research, while the exact mechanisms are not fully understood. There is evidence to suggest that the gut microbiota can influence the immune system and potentially play a role in the development or exacerbation of autoimmune conditions. Literature revealed alterations in the gut microbiome may induce GO by impacting the levels of thyroid-stimulating hormone receptor antibodies (TSH Ab) and fostering an imbalance in Th17/Treg (reference 1,2). Experimental findings in GD/GO mouse models revealed a reduction in disease severity with oral antibiotic vancomycin (reference 3). However, fecal microbiota transplantation from GD/GO patients exacerbated the condition (reference 2,3). Additionally, the composition of microbiota could be impacted by various factors, such as glucocorticoid, antithyroid drugs and immunosuppressants. GD/GO patients exhibit distinct differences in microbiota composition compared to healthy controls (reference 3). In my opinion, it's important to establish the link between gut microbiota and GD/GO, while significant differences in gut microbiota composition caused by various factors could explain variations in study results and clinical application. We have revised the manuscript and added the new paragraph as follows (page 7 of 15, line 270): Gut Microbiome in GO.
reference 1: Masetti, G., Moshkelgosha, S., Köhling, H. L., Covelli, D., Banga, J. P., Berchner-Pfannschmidt, U., et al. (2018). Gut Microbiota in Experimental Murine Model of Graves' Orbitopathy Established in Different Environments may Modulate Clinical Presentation of Disease. Microbiome 6, 97. doi: 10.1186/s40168-018-0478-4
reference 2: Su, X., Yin, X., Liu, Y., Yan, X., Zhang, S., Wang, X., et al. (2020). Gut Dysbiosis Contributes to the Imbalance of Treg and Th17 Cells in Graves' Disease Patients by Propionic Acid. J. Clin. Endocrinol. Metab. 105 (11), 3526–3547. doi: 10.1210/clinem/dgaa511
reference 3: Moshkelgosha, S., Verhasselt, H. L., Masetti, G., Covelli, D., Biscarini, F., Horstmann, M., et al. (2021). Modulating Gut Microbiota in a Mouse Model of Graves' Orbitopathy and its Impact on Induced Disease. Microbiome 9, 45. doi: 10.1186/s40168-020-00952-4
Response to Comments on the Quality of English Language
Point: English language fine. No issues detected
Response: Thank you for taking the time to review our manuscript.

Reviewer 3 Report
Comments and Suggestions for Authors
This is an interesting review, which is densely written, to the point of questionable grammatical statements. A lot is packed in here, but not much of it makes sense .... until you get to the conclusion.
Table 1 nd figure 2 are amazing, and they should be at the start of the manuscript, not the end. They set-up the manuscript, and even to some extent, even replace some of the text therein. If it is in the table, then there is no need to repeat it in the text!
Figure 1 - needs to be at the start, with arrows indication where each of the treatments fit into the scheme. as a section between introduction and the individual treatments. Then, a section entitled individual treatments -with the first sentence "See table 1 for summary" or something to that effect.
Then, fix the rest of the manuscript - grammatical issues and oddities
Finally, the conclusions - as it, it is quite worthless. the function of a conclusion is to put forth the main message of the manuscript AND THEN future directions.
Comments on the Quality of English LanguageThere are several instances of impenetrable english.
e.g.: line 62: Further what? Super activated or more cells are activated?
lines 69 - move CD90 to end of line ( after the word "marker". Then proof the sentence as it is missing a few "the's" and that's"
Why are random phrases italicized (lines 80-83)or bolded (lines 199-200 - unpublished study)?
sequela? line 271lines 323-325 - poorly constructed sentence
Please use passive voice - not "our study...." etc... e.g.: Lines 106 and 117
Lines 163/164 - please subscript hydrogen peroxide
Please don't cite unpublished studies as facts. E.g lines 302-304
Table 1: what does [reference] refer to?
Author Response
Response to Reviewer 3 Comments
Thank you for taking the time to review our manuscript and offer valuable feedback and comments. On behalf of all the authors of this manuscript (manuscript number ijms-2872935), I submit our responses to the reviewers’ comments and suggestions. The revised parts are highlighted in yellow for your convenience.
Point-by-point response to Comments and Suggestions for Authors
Comment 1: Table 1 nd figure 2 are amazing, and they should be at the start of the manuscript, not the end. They set-up the manuscript, and even to some extent, even replace some of the text therein. If it is in the table, then there is no need to repeat it in the text!
Response 1: Thank you for your suggestion. We have revised our introduction as follows (page 1 of 15, line 42): This review summarizes the current understanding of extracellular matrix proteins and associated tissue remodeling in the pathogenesis and potential mediators for the treatment of GO, and summarized in the figure 2 and table 1, respectively.
Comment 2: Figure 1 - needs to be at the start, with arrows indication where each of the treatments fit into the scheme. as a section between introduction and the individual treatments. Then, a section entitled individual treatments -with the first sentence "See table 1 for summary" or something to that effect.
Response 2: Thank you for your suggestion. We have revised figure 1 with arrows indicated and the figure 1 legend as follows (page 2 of 15, line 47): Figure 1 Computed tomography images in a patient with Graves' ophthalmopathy (GO). (A) Exposure keratopathy with eyeball rupture (as solid arrows) due to severe extraocular muscle hypertrophy and adipogenesis in the right eye. (B) Extraocular muscle hypertrophy in the left eye. (C,D) The patient received an evisceration with an implant (as dashed arrows) of the right eye, and orbital decompression of the left eye. The pre-operative image and post-operative image were demonstrated in figure C and D, respectively.
We have revised the introduction part as follows (page 1 of 15, line 42): This review summarizes the current understanding of extracellular matrix proteins and associated tissue remodeling in the pathogenesis and potential mediators for the treatment of GO, and summarized in the figure 2 and table 1, respectively.
Comment 3: Then, fix the rest of the manuscript - grammatical issues and oddities
Response 3: Thank you for your crucial observation. We have thoroughly reviewed our manuscript, addressing and rectifying grammatical issues.
Comment 4: Finally, the conclusions - as it, it is quite worthless. the function of a conclusion is to put forth the main message of the manuscript AND THEN future directions.
Response 4: Thank you for your suggestion. We have revised the concluding remarks as follows (page 10 of 15, line 415): In this review, we have demonstrated a picture of the crucial role and emerging issues related to ECM proteins in GO pathogenesis. To understand the mechanisms of ECM proteins dysregulation in GO, it is important to elucidate the molecular basis of the excessive proliferation and ECM production by GO orbital fibroblasts and the involvement of certain signaling pathways, leading to tissue expansion/remodeling and fibrosis in GO.
Current pharmacological agents are systemic steroids and immunomodulating agents, which may reduce inflammation but have limited effects on the long-term sequela. While teprotumumab, a monoclonal antibody targeting the insulin-like growth factor-1 receptor (IGF-1R), has shown remarkable success in managing active proptosis in GO, its cost, adverse effects, and infusion requirements emphasize the need for additional therapeutic options. Multiple antioxidants and antifibrotic agents, such as Pirfenidone, hold promise as potential therapies or prophylactic agents, while warranting long-term and large-scaled clinical trials. The identification of novel biomarkers, including antagonists in ER stress, PPARγ pathway, small molecules of histone and non-coding RNA, provide a more comprehensive understanding of the disease pathways. The ideal scenario would involve the oral or subcutaneous delivery of these novel therapies with minimal side effects. Broadening the concepts of the ECM and tissue fibrosis will help in the development of new therapeutic targets in GO.
Response to Comments on the Quality of English Language
Moderate editing of English language required.
There are several instances of impenetrable english.
Point 1: e.g.: line 62: Further what? Super activated or more cells are activated?
Response 1: We have revised our manuscript as follows (page 2 of 15, line 65): Activated orbital fibroblasts differentiation into fat storing adipocytes through adipogenesis and myofibroblasts trans-differentiation and ECM molecules secretion would be discussed in the following sections.
Point 2: lines 69 - move CD90 to end of line ( after the word "marker". Then proof the sentence as it is missing a few "the's" and that's"
Response 2: We have revised our manuscript as follows (page 2 of 15, line 72): Thy-1(+) or Thy-1(-). Thy-1 is a classical T lymphocyte marker (CD90) expressed on fibroblasts.
Point 3: Why are random phrases italicized (lines 80-83)or bolded (lines 199-200 - unpublished study)?
Response 3: Thank you for your observation.
We have revised the typing error and the italics as follows (page 3 of 15, line 77): The PI3K pathway and lipid mediators induce the activation of peroxisome proliferator-activated receptor γ (PPARγ) and enhance the adipogenesis. PPARγ is a nuclear receptor functioning as a transcription factor in lipid uptake and adipogenesis. Kumar et al demonstrated several adipogenesis-associated mRNA in orbital adipose tissues from GO patients, including leptin, adiponectin, PPARγ and preadipocyte factor-1, were elevated in parallel with TSH receptors. After IGF-1 stimulation, PPARγ was up-regulated, and PI3K pathway was activated. Thus, the inhibitors of PPARγ may bring a potential therapeutic target for GO. Notably, some type 2 diabetes medication, rosiglitazone or piogliatozone, which would activate PPARγ, might deteriorate GO.
We have revised the bolded error as follows (page 5 of 15, line 200): Our recent study (unpublished data) further demonstrates that ER protein TXNDC5 play an important role in TGF-β1-induced myofibroblast trans-differentiation and ECM protein in GO orbital fibroblasts.
Point 4: sequela? line 271lines 323-325 - poorly constructed sentence
Response 4: We have revised the manuscript in line 271 as follows (page 7 of 15, line 291): Current pharmacological agents are systemic steroids and immunomodulating agents, which may reduce inflammation but have limited effects on the long-term tissue remodeling.
We have revised the manuscript in line 323-325 as follows (page 8 of 15, line 342): The trial demonstrated noteworthy enhancements in both quality of life and ocular involvement, coupled with a reduced progression rate in patients with mild GO following a 6-month selenium supplementation. However, this recommendation was not endorsed by the American Thyroid Association (ATA) and the European Thyroid Association (ETA).
Point 5: Please use passive voice - not "our study...." etc... e.g.: Lines 106 and 117
Response 5: We have revised the manuscript as follows (page 3 of 15, line 108): In our prior study, the increased phosphorylation of p38 and JNK (but not ERK) after the induction of TGF-β1 was demonstrated, as well as the expression of connective tissue growth factor (CTGF), α-smooth muscle actin (α-SMA), fibronectin, tissue inhibitors of metalloproteinases 1 (TIMP-1) and TIMP-3. The levels of matrix metalloproteinases 2 (MMP-2) and MMP-9 were inhibited.
We have revised the manuscript as follows (page 3 of 15, line 119): In our previous study, the expression of CTGF, fibronectin, and α-SMA in Graves’ orbital fibroblasts were stimulated after the induction of TGF-β1.
Point 6: Lines 163/164 - please subscript hydrogen peroxide
Response 6: We have revised the manuscript as follows (page 4 of 15, line 165): hydrogen peroxide (H2O2).
Point 7: Please don't cite unpublished studies as facts. E.g lines 302-304
Response 7: Thank you for your observation. We have deleted the unpublished studies and revised the manuscript and table 1.
Point 8: Table 1: what does [reference] refer to?
Response 8: The [reference] in table 1 was corresponding with references in the manuscript. Thank you for your observation.
